# Unsupervised Prompt Learning with Few-shot Examples for Answering Objective Questions

## Abstract

*Large language models* (LLMs) have been highly successful on diverse tasks, while some applications require specializing general purpose LLMs to meet stricter accuracy or latency targets; here we focus on objective question answering, an important real-world setting in which a nontrivial subset benefits from such specialization. Most existing methods require parameter retraining or human supervision, both entailing high computational and data collection burdens. To handle these challenges, a direct approach is to generate "high-confidence" data from unsupervised downstream tasks and use them for prompt learning or in-context learning to efficiently refine pseudo-supervision. We consider combining the two approaches for better performance; however, a naive strategy that learns the prompt first and selects pseudo-supervised examples only at inference creates a mismatch between prompt learning and usage. In this paper, we propose *unsupervised few-shot prompt learning* (UFPL), which jointly learns the prompt and refines the overall pseudo-supervision. The learning objective aligns prompt training with usage by requiring the learned prompt to produce consistent answers when pseudo-supervised data from the downstream task are used as in-context examples. We optimize the prompt by translating gradient signals into textual critiques, which serve as feedback to iteratively refine the prompt and the pseudo supervision. Theoretical analysis in a simplified classification setting shows that the algorithm implicitly introduces a regularization, supporting its design. Empirical results on diverse benchmarks and a real world molecule optimization task show the effectiveness of our approach.

## 1 Introduction

Large language models have shown impressive performance on various real-world tasks (Brown et al., 2020; Achiam et al., 2023). While broadly competent across downstream tasks, applications with strict accuracy, latency, or safety targets, often benefit from *specialization of general purpose LLMs* (Shin et al., 2020; Ouyang et al., 2022). We focus on *objective question answering*, an important real-world setting in which each query has a single verifiable answer. This setting encompasses many practical tasks that can benefit from specialization, including clinical question answering (Singhal et al., 2025), instruction following (Wei et al., 2021), and so on.

Existing methods for LLM specialization mainly involve fine-tuning, prompt optimization, and in-context learning. Parameter efficient fine-tuning, such as LoRA (Hu et al., 2021), updates a small subset of parameters to specialize the LLM. By contrast, prompt learning (Sun et al., 2022; Zhou et al., 2022) and in-context learning (Brown et al., 2020; Liu et al., 2022) adapt behavior by learning input prompts or providing few-shot examples, leaving the base LLM unchanged. Although these approaches share the same goal, fine-tuning produces additional model variants that raise storage overhead, and when tasks arrive as a stream or change frequently, repeatedly loading and switching variants adds latency and operational complexity. In such cases, retaining a general purpose LLM with task appropriate prompts and data-specific few-shot examples can be preferable (Vu et al., 2022).

To facilitate efficient specialization of general purpose LLMs, we adopt prompt learning and in-context learning, which optimize only the input, to improve performance on downstream tasks. These methods typically require human supervision to induce prompts or to supply few-shot examples; however, human feedback is costly and time-consuming to collect. Fortunately, in many real-world tasks, LLMs can annotate text datasets with quality that matches or exceeds human annotators, motivating

**Figure 1:** Pipeline comparison of UFPL and Naive Combination.

the use of unlabeled data with LLMs. For example, GPT-3 has been reported to surpass human performance on text classification (Gilardi et al., 2023),and more broadly LLMs perform strongly on natural language classification benchmarks (Chong et al., 2022). Motivated by these observations, we study an unsupervised specialization setting that uses LLM generated pseudo-supervision.

It is nontrivial to combine prompt learning and in-context learning to produce refined pseudo-supervision for specialization in an unsupervised setting. A naive approach first identifies "high confidence" pseudo-supervised data (e.g., via *chain-of-thought* (CoT) reasoning (Wei et al., 2022)), and then either optimizes the prompt using these data (Diao et al., 2023; Sun et al., 2022) and uses them as few shot examples for in context prediction (Wan et al., 2023b; Guo et al., 2024; Li et al., 2024). However, this decoupled design is problematic: it uses few-shot examples only during inference rather than during prompt learning, creating a mismatch between how the prompt is learned and how it is used at inference, namely task shift.

In this paper, we propose *unsupervised few-shot prompt learning* (UFPL), which jointly optimizes the prompt and the pseudo-supervision that determines the few-shot examples, ensuring consistency between prompt training and usage, as illustrated in Figure 1. Specifically, we iteratively identify "high-confidence" pseudo-labeled data and, using these data, align prompt training with usage by requiring the learned prompt to produce consistent answers when pseudo-supervised data from the downstream task are used as in-context examples. We use TextGrad (Yuksekgonul et al., 2024) to optimize the prompt via gradient based updates driven by textual feedback, akin to gradient descent, yielding an approach applicable to both open source and black box models. Theoretical analysis shows that, in the simplified setting of classification, the proposed algorithm implicitly introduces a regularization and the refined output exhibits a cluster structure that helps alleviate the overfitting issue. We evaluate UFPL with other contenders on several benchmark datasets and a real-world molecule optimization task. Experimental results show that UFPL produces high-quality refined generations without human supervision, regardless of model scale.

## 2 RELATED WORK

**Prompt Learning and In-context Learning.** Recent advances in prompt learning have developed more systematic methods for prompt design and optimization. Pioneering work such as BBT learns prompts using derivative free optimization techniques like evolutionary algorithms (Sun et al., 2022). BDPL employs policy gradient algorithms to optimize the prompt (Diao et al., 2023). Besides, gradient based prompt learning methods were also proposed, including ProTeGi (Pryzant et al., 2023), TextGrad (Yuksekgonul et al., 2024), and GREATER (Das et al., 2024). Typically, these methods still require human supervision to optimize the prompt. Chain-of-Thought prompting introduces unsupervised step-by-step reasoning for complex tasks, enabling LLMs to decompose a hard problem into intermediate steps and solve them sequentially (Wei et al., 2022). Subsequent work improves reliability by sampling multiple reasoning paths and selecting the most consistent answer (Wang et al., 2022), and extends CoT by exploring multiple reasoning branches in a tree like structure for complex problem solving (Yao et al., 2023). Although they do not rely on downstream supervision, these methods solve problems individually and cannot leverage other data available in the downstream task.

In-context learning conditions a frozen LLM on task descriptions and few-shot examples at inference time, enabling rapid adaptation without parameters updates (Brown et al., 2020). Subsequent analyses investigated what makes ICL effective, highlighting the roles of label space cues, surface form overlap, and example quality (Min et al., 2022; Liu et al., 2022). A parallel line studied practical levers for ICL, including calibrated decoding to mitigate majority label bias (Zhao et al., 2021) and ordering strategies that reduce sensitivity to example permutations (Lu et al., 2022). Our work leverages these

insights but targets unsupervised specialization: we jointly learn prompts and pseudo-supervised few-shot examples so that prompt training is aligned with prompt use at inference.

Recent seminal works have explored using in-context prompting for LLM specialization without retraining model parameters (Wan et al., 2023a;b; Li et al., 2024). These methods first identify "high-confidence" pseudo-supervised data using carefully designed scoring functions, and then leverage the selected data as in-context examples to guide final predictions. We jointly learn the task appropriate prompt and the pseudo-supervision, making the prompt learning and using stages more consistent.

**Self-supervised Fine-tuning.** In an unsupervised setting, self-supervised fine-tuning methods first use the LLM to generate pseudo-supervised data for a downstream task and then adapt the LLM via parameter-efficient fine-tuning. For instance, LMSI employs CoT prompting (Wei et al., 2022) to generate high-quality labels for unlabeled datasets, which were then used to optimize the model (Huang et al., 2023). LLMRefine employs a fine-grained feedback model to identify defects in outputs and guide iterative refinements, optimizing performance during inference without additional training (Xu et al., 2024). Similarly, SALMON retrieves high-quality samples relevant to the downstream task and used them as in-context examples to generate additional samples, which were then iteratively employed to fine-tune the LLM (Sun et al., 2024). ISARA is an improved self-refinement methods without human-crafted instructions and labeled rewards (Guo et al., 2024).

## 3 OUR APPROACH

In this section, we begin by introducing the notations, then describe the UFPL algorithm in detail, and finally provide a theoretical analysis of its properties in a simplified setting of classification.

### 3.1 NOTATIONS

In this part, we introduce the notations. Let $\mathbf{x}_l \in \mathcal{X}$ be the $l$-th query in the unsupervised dataset of size $n$, where $\mathcal{X}$ is the textual space. We denote by $\mathbf{z} \in \mathcal{X}$ the prompt and $\mathbf{z}_0$ be the initial prompt. We define the generation function as $\text{LLM}_{\text{gen}}(\cdot, \cdot, \cdot) : (\mathbf{x}, \mathbf{z}, D) \mapsto \mathbf{y}$, where $\mathbf{x}$ is the input, $\mathbf{y} \in \mathcal{X}$ is the answer in textual space, $\mathbf{z}$ is the prompt, and $D = \{(\mathbf{x}_k, \widehat{y}_k)\}_{k=1}^K$ is a set of $K$ pseudo-supervised examples drawn from the downstream task. We denote inference by $\text{LLM}_{\text{gen}}(\mathbf{x}, \mathbf{z}, D)$. When $D = \emptyset$, e.g., $\text{LLM}_{\text{gen}}(\mathbf{x}, \mathbf{z}_0, \emptyset)$, the model predicts with a default prompt and no examples.

To learn the prompt, following TextGrad (Yuksekgonul et al., 2024), we define a text valued loss $L(\cdot \mid \cdot, \cdot, \cdot) : (\mathbf{z} \mid \mathbf{x}, \mathbf{y}, \mathbf{y}) \mapsto \mathbf{p}$, where $\mathbf{p} \in \mathcal{X}$ denotes a textual loss signal such as a prediction consistency critique. For example, $L(\mathbf{z} \mid \mathbf{x}, \widehat{y}, y)$ returns an LLM generated critique that evaluates how well the pseudo-supervision $\widehat{y}$, produced using $\mathbf{z}$, addresses $\mathbf{x}$ relative to the underlying supervision $y$. For notational simplicity, when the context is clear we write $L(\mathbf{z})$ in place of $L(\mathbf{z} \mid \mathbf{x}, \widehat{y}, y)$.

Next, we define a prompting operator $\text{LLM}_{\text{grad}}(\cdot) : \mathbf{p} \mapsto \mathbf{g}$ that maps a textual loss $\mathbf{p}$ to a textual gradient $g \in \mathcal{X}$, i.e., a signal indicating a direction for improvement. Concretely, given $L(\mathbf{z})$, we obtain update instructions by

$$\frac{\partial L}{\partial \mathbf{z}} := \text{LLM}_{\text{grad}}(L(\mathbf{z})) \tag{1}$$

Finally, we define an update operator $\text{LLM}_{\text{update}}(\cdot, \cdot) : (\mathbf{z}, \mathbf{g}) \mapsto \mathbf{z}$ that applies a textual gradient to produce a refined prompt, in analogy to a gradient step:

$$\mathbf{z}_{\text{new}} = \text{LLM}_{\text{update}}(\mathbf{z}_{\text{old}}, \frac{\partial L}{\partial \mathbf{z}}). \tag{2}$$

### 3.2 UNSUPERVISED FEW-SHOT PROMPT LEARNING

In this part, we present the proposed UFPL algorithm, which jointly optimizes the prompt and refines the pseudo-supervision for the downstream task in an *iterative* manner.

Since the downstream task is unsupervised, we first identify "high-confidence" pseudo-supervised data as initialization. Following prior work (Huang et al., 2023), we use self-consistency CoT (Wang et al., 2022) both to select these data and to estimate the confidence of pseudo-supervised data. Specifically,

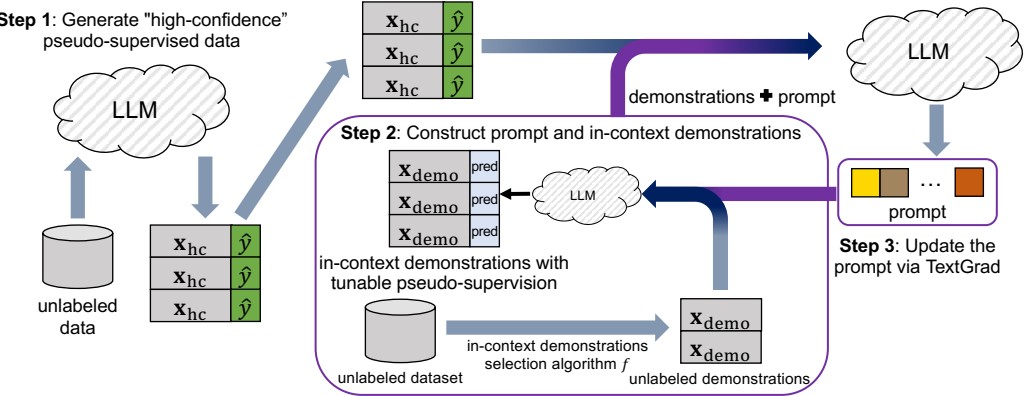

We iteratively **Step 1**: identify "high-confidence" pseudo-supervised data and, for each instance, construct few shot examples by selecting a set of data from the downstream task with a selection algorithm. **Step 2**: assign to each selected data pseudo-supervision generated by the LLM conditioned on the current prompt and its few-shot examples. **Step 3**: jointly refine the pseudo supervision and learn the prompt on the high confidence set according to equation 5.

**Figure 2:** An illustration of the UFPL algorithm.

for each input $\mathbf{x}_l$, we perform multiple-path decoding with temperature $T > 0$, generating $m$ reasoning paths with prompt $\mathbf{z}$ and the corresponding answers $\{y_{l_1}, \ldots, y_{l_m}\}$. We then apply majority voting (self-consistency) to select the most consistent answer $\widehat{y}_l$ and define its confidence as:

$$c_l = \frac{1}{m} \sum_{j=1}^{m} \mathbb{1}(y_{l_j} = \widehat{y}_l). \tag{3}$$

To specialize an LLM for a given downstream task, a straightforward approach is to learn the prompt based on these "high-confidence" pseudo-supervised data, following the principal idea from (Wan et al., 2023b; Guo et al., 2024; Li et al., 2024). For example, we optimize the following objective:

$$\arg\min_{\mathbf{z} \in \mathcal{Z}} \sum_{l=1}^{n} \mathbb{1}[c_l \geq \gamma] \cdot L(\mathbf{z} \mid \mathbf{x}_l, \mathrm{LLM}_{\mathrm{gen}}(\mathbf{x}_l, \mathbf{z}, \emptyset), \mathrm{LLM}_{\mathrm{gen}}(\mathbf{x}_l, \mathbf{z}_0, \emptyset)), \tag{4}$$

where $\mathbb{1}[\cdot]$ is the indicator function and $\gamma \in [0, 1]$ is a threshold for selecting "high-confidence" pseudo-supervised data in the downstream task.

However, prompt learning based on "high-confidence" pseudo-supervised data, as in Eqn. equation 4, creates a mismatch between prompt training and using phases because the objective does not consider the examples used at inference. Since inference uses $\mathrm{LLM}_{\mathrm{gen}}(\mathbf{x}, \mathbf{z}, D)$, we jointly learn the prompt and refine the task's pseudo-supervision, with the refined pseudo-labeled data serving as few-shot examples at inference. Therefore, we define the objective for unsupervised few-shot prompt learning:

$$L_{\mathrm{m}}(\mathbf{z}) = \sum_{l=1}^{n} \mathbb{1}[c_l \geq \gamma] \cdot L(\mathbf{z} \mid \mathbf{x}_l, \mathrm{LLM}_{\mathrm{gen}}(\mathbf{x}_l, \mathbf{z}, D_l), \mathrm{LLM}_{\mathrm{gen}}(\mathbf{x}_l, \mathbf{z}_0, \emptyset)), \tag{5}$$

where $D_l$ denotes the set of in-context examples for input $\mathbf{x}_l$ selected by algorithm $f(\mathbf{x}_l; \mathbf{z})$, the pseudo-supervision of these examples is also generated with prompt $\mathbf{z}$ and their examples. Specifically, $f(\mathbf{x}_l, \mathbf{z})$ returns a set of pseudo-supervised examples drawn from the downstream task:

$$D_l = \{(\mathbf{x}_k, \mathrm{LLM}_{\mathrm{gen}}(\mathbf{x}_k, \mathbf{z}, \mathcal{D}_k)) | \mathbf{x}_k \in S_l\}_{k=1}^{K},$$

where $\mathrm{LLM}_{\mathrm{gen}}(\mathbf{x}_k, \mathbf{z}, \mathcal{D}_k)$ is the pseudo-supervision of $\mathbf{x}_k$, guided by both $\mathbf{z}$ and $\mathcal{D}_k$.

For few-shot examples selection, following seminal works on in-context example selection (Liu et al., 2022; Min et al., 2022), we choose, for each input $\mathbf{x}_l$, its $K$ nearest neighbors as the in-context examples, denoted by $S_l$:

$$S_l = \arg\min_{\{k_j\}_{j=1}^{K} \subset \{1, \ldots, n\}} \sum_{j=1}^{K} d(\mathbf{x}_l, \mathbf{x}_{k_j}), \tag{6}$$

where $d(\cdot, \cdot)$ denotes a distance between two inputs; for example, we use $d(\mathbf{x}_l, \mathbf{x}_k) = \|\theta(\mathbf{x}_l) - \theta(\mathbf{x}_k)\|_2$ with $\theta(\cdot)$ a sentence encoder (Liu et al., 2022). In addition, to mitigate majority label bias in the in-context examples, we adopt the plug-in de-biasing method of (Zhao et al., 2021).

We illustrate the proposed UFPL algorithm in Figure 2 and provide pseudo-code in Algorithm 1. Our algorithm proceeds iteratively and can be terminated early depending on the time and cost constraints of the downstream task.

### 3.3 THEORETICAL ANALYSIS

In this section, we present theoretical insights for the UFPL algorithm. The included theorem is standard and intended solely to support the approach, not to claim a theoretical contribution.

Informally, the analysis shows that UFPL refines generation by encouraging the pseudo supervision to form a clustered structure in the output space. In the simplified multi-class classification setting, UFPL promotes a multi-manifold structure in which each class occupies a disjoint convex region. Consequently, queries with similar semantics are encouraged to receive the same refined label, helping mitigate overfitting and improve generalization (Chapelle et al., 2006; Belkin et al., 2006).

Recent seminal works have shown that ICL can be interpreted as a form of implicit *empirical risk minimization* (ERM) (Min et al., 2022; Xie et al., 2022; Bai et al., 2023). We begin by recalling the following lemma from Bai et al. (2023).

**Lemma 1** (Corollary G.1 in (Bai et al., 2023))**.** *For any transformer with layer $L \geq 1$, under the same setting as Theorem G.1 in (Bai et al., 2023), the $(2L)$-layer transformer $TF_\theta$ there approximates the true gradient descent trajectory $\{\mathbf{w}_{\text{GD}}^\ell\}_{\ell \geq 0}$: For the intermediate iterates $\{\widehat{\mathbf{w}}^\ell\}_{\ell \in [L]}$ considered therein, we have*

$$\|\widehat{\mathbf{w}}^\ell - \mathbf{w}_{\text{GD}}^\ell\|_2 \leq L_f^{-1}(1 + \eta L_f)^\ell \varepsilon,$$

*where $L_f = \sup_{\mathbf{w} \in \mathcal{W}} \|\nabla^2 \widehat{L}_N(\mathbf{w})\|_{\text{op}}$ denotes the smoothness of $\widehat{L}_N$ within $\mathcal{W}$.*

Lemma 1 shows that, under the mild technical assumptions in Bai et al. (2023), a $(2L)$-layer transformer *approximates the true gradient-descent trajectory*, with the intermediate iterates closely tracking gradient descent in context. Viewed through this lens, ICL performs an ERM like procedure over the provided few-shot examples: the model instantiates an implicit classifier conditioned on these examples and applies it to new data. Therefore, UFPL enforces consistency of the refined pseudo-supervision under such examples-conditioned ERM: for any pseudo-supervised data, conditioning on pseudo-supervised examples induces (via ICL) an implicit classifier that is applied to that data, and the resulting prediction is encouraged to agree with its pseudo supervision.

Based on this observation, we make the following assumption about the refined outputs of UFPL.

**Assumption 1** (Leave-one-out correctness)**.** Consider a multi-class classification task (e.g., multiple-choice QA). Let $S = \{(x_i, y_i)\}_{i=1}^n$ and, for each $i$, let $S^{(-i)} := S \setminus \{(x_i, y_i)\}$. Denote by $f^{(-i)} : \mathbb{R}^d \to \mathbb{R}^K$ the score function returned by the (demonstration-conditioned) ERM (via ICL) trained on $S^{(-i)}$, and define the induced classifier $h^{(-i)}(x) := \arg\max_{k \in \{1,\dots,K\}} f_k^{(-i)}(x)$. We assume *leave-one-out correctness*: for every $i \in \{1, \dots, n\}$,

$$h^{(-i)}(x_i) = y_i.$$

**Assumption 2** (Uniform stability)**.** Let $\mathcal{A}$ be the (demonstration-conditioned) regularized ERM procedure that maps a sample to a predictor; denote by $f_S : \mathbb{R}^d \to \mathbb{R}^K$ and $f_{S^{(-i)}}$ the score functions returned by $\mathcal{A}$ on $S$ and $S^{(-i)}$, respectively. Let $\ell : \mathbb{R}^K \times \{1, \dots, K\} \to \mathbb{R}_+$ be the per-example loss (e.g., cross-entropy or hinge), and assume $\ell$ is $L$-Lipschitz in its score argument.

We say $\mathcal{A}$ is *uniformly stable* with parameter $\beta > 0$ if for all $i \in \{1, \dots, n\}$ and all $z = (x, y)$,

$$\left| \ell\big(f_S(x), y\big) - \ell\big(f_{S^{(-i)}}(x), y\big) \right| \leq \frac{\beta}{n}.$$

**Theorem 1.** *Under Assumptions 1 and 2, let $S = \{(x_i, \widetilde{y}_i)\}_{i=1}^n$ be the refined pseudo supervision produced by UFPL. Let $f_S : \mathbb{R}^d \to \mathbb{R}^K$ be the ERM score and $h_S(x) = \arg\max_{k \leq K} f_{S,k}(x)$ the induced classifier. Then there exists $\gamma > 0$ such that for all $i$,*

$$f_{S,\widetilde{y}_i}(x_i) - \max_{j \neq \widetilde{y}_i} f_{S,j}(x_i) \geq \gamma.$$

---

**Algorithm 1** Unsupervised Few-shot Prompt Learning (UFPL)

---

1: Set total number of iterations $T$, number of in-context demonstrations $K$, total number of sampling $m$ for confidence estimation, and confidence threshold $\gamma$.
2: **for** $t = 1$ **to** $T$ **do**
3:     **Stochastic sampling:** Sample a mini-batch of data from the downstream task
4:     **Confidence estimation:** Estimate the confidence by equation 3 with $\mathbf{z}^{(t)}$
5:     **Compute loss:** Compute loss by equation 5 and generate gradient by equation 1
6:     **Update prompt:** $\mathbf{z}^{(t+1)} = \text{LLM}_{\text{update}}(\mathbf{z}^{(t)}, \frac{\partial L}{\partial \mathbf{z}^{(t)}})$
7:     **Refine output:** $\forall l \in [n], \widehat{y}_l = \text{LLM}_{\text{gen}}(\mathbf{x}_l, \mathbf{z}^{(t+1)}, D_l)$
8: **end for**

---

*If, in addition, $f_S$ is L-Lipschitz in $x$, set $r := \gamma/(3L)$. Then for any point $x$ and any training point $x_i$, if $\|x - x_i\|_2 < r$ then $h_S(x) = \widetilde{y}_i$. In particular, if two training points $x_i, x_j$ can be connected by a chain $x_i = z_0, z_1, \ldots, z_m = x_j$ with $\|z_{t+1} - z_t\|_2 < r$ for all $t$, then $\widetilde{y}_i = \widetilde{y}_j$.*

Theorem 1 shows that the pseudo supervision refined by UFPL exhibits a low dimensional, cluster aligned geometry consistent with the clustering induced by graph Laplacian minimization, indicating that UFPL imposes an implicit regularization in the output space, which helps mitigate overfitting and improve generalization. Detailed proofs are deferred to the **Appendix B**.

## 4 EXPERIMENTS

In this section, we evaluate UFPL alongside six contenders on a range of benchmarks. We then conduct ablation studies to quantify runtime and cost and to assess the contribution of each component. Finally, we test the proposed algorithm on a real-world molecular optimization task.

### 4.1 EXPERIMENTAL SETUP

**Datasets.** We evaluate UFPL on a comprehensive suite of benchmarks spanning question answering, reasoning, mathematics, and natural language understanding. The evaluation covers the MMLU benchmark (Hendrycks et al., 2021) (AST, HSCS, HSM, CMath, CCS, CMed, MAN, MAR, and RND); GPQA (Rein et al., 2024); SimpleQA (Wei et al., 2024); TruthfulQA (Lin et al., 2022); GSM8k (Cobbe et al., 2021); Hellaswag (Zellers et al., 2019), and BBH dataset (Suzgun et al., 2023).

We also evaluate UFPL on a real-world molecular optimization task using the DOCKSTRING dataset (García-Ortegón et al., 2022). Each molecule is represented as a SMILES string (Yuksekgonul et al., 2024), and the learning problem is to generate an improved version that surpasses the original in terms of important chemical properties, specifically the Vina score, which reflects binding affinity, and the QED score, which measures drug-likeness (Trott & Olson, 2010).

**Contenders.** We compare UFPL with six contenders: two baselines that directly generate answers (with and without chain-of-thought reasoning), two in-context learning algorithms, and two strong prompt learning contenders with pseudo-supervision in the downstream tasks. Specifically,

**Direct** prompts the LLM with a default prompt to produce answers. **Auto-CoT** (Zhang et al., 2022) induces intermediate reasoning at inference, encouraging a chain-of-thought before the final answer.

Using the same mechanism as UFPL to select "high-confidence" pseudo supervised examples, we evaluate **ICL** (Liu et al., 2022), which uses these examples as few-shot examples to predict the remaining unlabeled data, and **USP** (Wan et al., 2023b), which scores and selects "high-confidence" data and then applies in-context learning for generation.

Following the self refinement strategy of (Huang et al., 2023), which performs prompt learning using "high-confidence" pseudo-supervised data, we intorduce **SR (BDPL)** (Diao et al., 2023), which optimizes prompts via policy gradient on pseudo-supervised data, and **SR (RLprompt)** (Deng et al., 2022), which uses a parameter efficient policy network to generate prompts conditioned on these examples. For both variants we use the default settings in their original papers.

For fairness, we apply the plug-in calibration method of (Zhao et al., 2021) to all contenders.

**Table 1:** Performance comparisons on benchmark datasets. We report the average accuracy (%) and standard deviation over 5 runs. The best average accuracy are in **bold**, and (↑ ·) indicates the improvement over Direct.

| Model | Dataset | Direct | ICL | Auto-CoT | USP | SR (BDPL) | SR (RLprompt) | PAPO |
|---|---|---|---|---|---|---|---|---|
| GPT-4o | GPQA | 47.4 ± 0.2 | 47.1 ± 0.5 | 47.9 ± 0.7 | 47.8 ± 0.8 | 48.3 ± 1.3 | 48.1 ± 1.9 | **49.7 ± 1.5** (↑2.3) |
| | SimpleQA | 38.2 ± 0.8 | 37.5 ± 1.2 | 38.9 ± 1.0 | 38.8 ± 1.1 | 38.1 ± 1.3 | 37.4 ± 1.1 | **39.6 ± 0.9** (↑1.4) |
| | TruthfulQA | 71.8 ± 0.5 | 72.3 ± 0.8 | 72.1 ± 1.2 | 72.2 ± 1.1 | 72.7 ± 1.7 | 72.3 ± 1.0 | **74.3 ± 1.2** (↑2.5) |
| | MMLU | 87.5 ± 0.3 | 87.7 ± 1.1 | 88.1 ± 1.3 | 88.0 ± 1.4 | 88.9 ± 2.1 | 89.2 ± 1.7 | **90.4 ± 1.9** (↑2.9) |
| | GSM8k | 93.9 ± 0.5 | 94.1 ± 0.8 | 94.3 ± 0.9 | 94.4 ± 1.0 | 94.5 ± 1.4 | 94.9 ± 1.5 | **95.7 ± 1.3** (↑1.8) |
| | HellaSwag | 94.7 ± 0.4 | 94.8 ± 0.6 | 95.1 ± 0.6 | 95.0 ± 0.7 | 95.7 ± 1.3 | 95.5 ± 0.8 | **96.3 ± 0.5** (↑1.6) |
| | BBH | 83.1 ± 0.8 | 83.4 ± 0.9 | 83.3 ± 1.1 | 83.4 ± 1.2 | 84.8 ± 1.5 | 85.1 ± 1.2 | **86.2 ± 0.8** (↑3.1) |
| Qwen3 | GPQA | 47.9 ± 1.3 | 47.3 ± 0.9 | 48.4 ± 0.5 | 48.3 ± 0.6 | 47.9 ± 1.0 | 47.5 ± 0.9 | **49.9 ± 0.6** (↑2.0) |
| | SimpleQA | 39.1 ± 0.7 | 38.7 ± 1.1 | 39.4 ± 0.9 | 39.5 ± 1.0 | 40.3 ± 1.5 | 40.7 ± 0.9 | **41.5 ± 1.1** (↑2.4) |
| | TruthfulQA | 72.9 ± 0.9 | 73.1 ± 1.0 | 73.3 ± 1.2 | 73.4 ± 1.1 | 74.4 ± 1.5 | 74.8 ± 1.2 | **75.3 ± 1.1** (↑2.4) |
| | MMLU | 85.3 ± 0.5 | 85.7 ± 1.0 | 86.1 ± 1.2 | 86.2 ± 1.2 | 86.6 ± 1.5 | 86.9 ± 1.6 | **88.1 ± 0.7** (↑2.8) |
| | GSM8k | 94.4 ± 1.8 | 94.6 ± 1.5 | 94.9 ± 1.6 | 94.8 ± 1.5 | 95.2 ± 1.4 | 94.5 ± 1.2 | **95.9 ± 1.3** (↑1.5) |
| | HellaSwag | 95.1 ± 0.9 | 95.3 ± 0.9 | 95.6 ± 0.8 | 95.5 ± 0.9 | 96.1 ± 1.1 | 96.2 ± 0.9 | **96.7 ± 0.8** (↑1.6) |
| | BBH | 87.5 ± 1.1 | 87.8 ± 1.2 | 88.0 ± 1.3 | 88.1 ± 1.3 | 88.5 ± 1.5 | 88.6 ± 1.3 | **88.9 ± 1.4** (↑1.4) |
| Llama | GPQA | 25.9 ± 0.4 | 26.7 ± 2.1 | 27.1 ± 1.8 | 27.2 ± 1.8 | 27.5 ± 2.2 | 27.8 ± 1.9 | **29.3 ± 1.7** (↑3.4) |
| | SimpleQA | 15.3 ± 0.9 | 15.6 ± 1.0 | 15.8 ± 1.1 | 15.9 ± 1.1 | 16.3 ± 1.5 | 16.5 ± 1.6 | **16.9 ± 2.1** (↑1.6) |
| | TruthfulQA | 31.9 ± 0.7 | 32.1 ± 0.9 | 32.3 ± 1.0 | 32.4 ± 1.0 | 32.7 ± 1.3 | 32.9 ± 1.4 | **33.1 ± 1.5** (↑1.2) |
| | MMLU | 49.1 ± 0.3 | 49.5 ± 1.7 | 49.8 ± 1.4 | 49.9 ± 1.5 | 50.7 ± 1.6 | 50.5 ± 1.5 | **52.8 ± 0.9** (↑3.7) |
| | GSM8k | 44.3 ± 0.4 | 45.1 ± 1.5 | 45.3 ± 1.9 | 45.4 ± 2.0 | 46.0 ± 1.9 | 46.3 ± 1.7 | **47.5 ± 1.3** (↑3.2) |
| | HellaSwag | 41.3 ± 0.6 | 41.8 ± 0.8 | 42.1 ± 0.9 | 42.2 ± 0.9 | 43.5 ± 1.1 | 44.0 ± 0.9 | **44.7 ± 0.6** (↑3.4) |
| | BBH | 37.6 ± 1.1 | 38.0 ± 1.2 | 38.6 ± 1.3 | 38.7 ± 1.3 | 39.5 ± 1.4 | 39.8 ± 1.3 | **40.3 ± 1.4** (↑2.7) |

**Implementation Details.** In all experiments, we employed GPT-4o [1], Qwen3-235B [2], and Llama-3.2-1B [3], spanning from black-box to open-source models, and from large-scale to small-scale LLMs. In the ablation studies, we employ GPT-4o and Llama-3.2-1B to evaluate the stability of the proposed algorithm. We set all termination $T = 3$. For both ICL and UFPL, the number of demonstrations is set to 5. The confidence threshold is fixed at $\gamma = 0.65$ for UFPL and all competing methods. Due to page limits, more implementation details on hyperparameters setting and prompts design are provided in **Appendix C**.

## 4.2 PERFORMANCE COMPARISON ON BENCHMARKS

In this section, we compare the UFPL algorithm with other contenders on benchmark datasets.

**Performance.** We report the mean accuracy and standard deviation of the refined answers produced by UFPL and other contenders in Table 1. The proposed UFPL algorithm consistently outperforms nearly all other methods across the evaluated datasets. Relative to Direct and Auto-CoT, UFPL achieves higher accuracy, indicating that leveraging unlabeled downstream data and optimizing the prompt can refine generation more effectively than simply eliciting chain-of-thought at inference. Moreover, UFPL surpasses ICL, USP, SR (BDPL), and SR (RLPrompt), underscoring the value of jointly refining the few-shot pseudo-supervised examples during prompt learning rather than selecting them only at inference. The gains persist across model scales and task types, and they hold under a common calibration scheme applied to all contenders, suggesting that the improvements arise from better alignment between prompt learning and usage rather than from evaluation artifacts.

**Runtime overhead and cost.** Next, we analyze the runtime overhead and cost of UFPL. Our method adds two sources of cost: (i) constructing a distance matrix over the unlabeled set at initialization for few-shot examples selection, and (ii) per round refinement of pseudo supervision coupled with prompt updates. The initialization is a one time cost that is amortized over subsequent rounds, while the per round refinement primarily adds selection and consistency checks on a small subset of examples.

We compare UFPL with four representative contenders: Direct, USP, SR (RLPrompt), and SFT-LoRA (Hu et al., 2021). The cost of SFT-LoRA is computed as the total GPU and CPU hours used for training on A100, multiplied by the on-demand hourly rate, and then amortized over the number

---

[1] https://platform.openai.com/docs/models/gpt-4o

[2] https://huggingface.co/Qwen/Qwen3-235B-A22B

[3] https://huggingface.co/meta-llama/Llama-3.2-1B

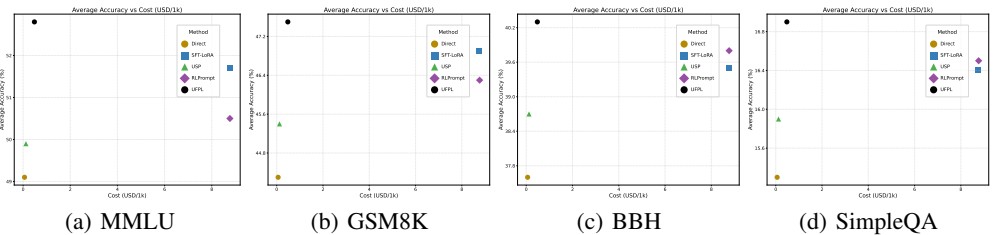

(a) MMLU      (b) GSM8K      (c) BBH      (d) SimpleQA

**Figure 3:** Average Performance v.s. Cost on Llama-3.2-1B.

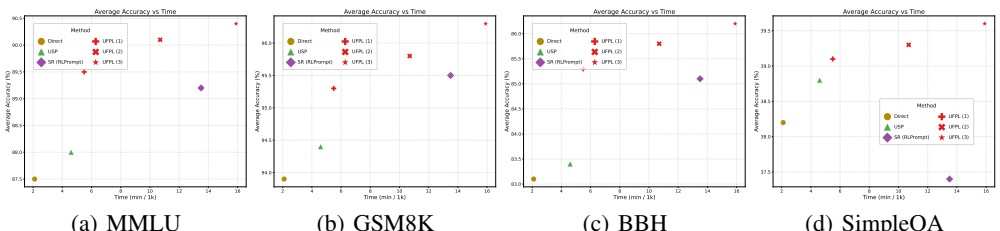

(a) MMLU      (b) GSM8K      (c) BBH      (d) SimpleQA

**Figure 4:** Average Performance v.s. Runtime Overhead on GPT-4o.

of inference tokens to obtain a per-1k-token cost. SFT-LoRA performs supervised fine-tuning on the same high confidence pseudo supervised data as USP and SR (RLPrompt).

We report the average accuracy and average cost of UFPL and competing method over ten runs with Llama-3.2-1B in Figure 3. The results show that UFPL attains higher accuracy with only a modest increase in cost, and this trend is consistent across datasets. Compared with Direct and USP, which have costs comparable to UFPL, UFPL shows better performance. Relative to SFT-Lora, our method achieves both higher accuracy and lower cost, indicating that fine-tuning a LLM can be more expensive than UFPL.

Figure 4 presents the average accuracy and run time over ten runs for all contenders on GPT-4o. Since UFPL is iterative, we report its performance for three runs. As shown in Figure 4, the second run of UFPL achieves a noticeable performance improvement with a reasonable runtime overhead.

**Effectiveness of each component.** To test the effectiveness of each component in UFPL, we introduce the following ablation baselines. Each variant isolates a specific aspect of the proposed pipeline, allowing us to understand its individual impact on performance across multiple datasets. The comparison methods are defined as follows:

**UFPL (textual conf.):** Replaces the CoT selection mechanism for initializing high-confidence data with a strategy based on the LLM's output likelihood scores.

**UFPL (w/o voting):** Replaces the confidence-based voting mechanism by estimating confidence from a single sampled output.

**UFPL (random sampling):** Replaces the K nearest neighbors selection with random sampling. Few-shot examples are selected randomly while ensuring class balance.

**UFPL (w RLPrompt):** Replaces the TextGrad optimization step in the proposed method with RLPrompt while keeping all other components unchanged.

**UFPL:** Represents the complete version of our proposed method, incorporating all components.

We report the results on GPT-4o in Table 2. We observe that all proposed components are beneficial for performance improvement: removing any single component consistently degrades performance, while using all components yields the best results across datasets.

**Beyond prompt optimization and the refinement of pseudo supervision.** In some real-world tasks, users may prefer a customized model instead of relying on refined generation for downstream applications. To support this, we use the refined pseudo-supervision and apply OpenAI's commercial

**Table 2:** Performance comparisons of UFPL variants on benchmark datasets. We report average accuracy (%) and standard deviation over 5 runs. The best results are in **bold**.

| Dataset | UFPL (textual conf.) | UFPL (w/o voting) | UFPL (random sampling) | UFPL (w. RLPrompt) | UFPL |
|---|---|---|---|---|---|
| MMLU | $87.9 \pm 1.5$ | $85.8 \pm 1.3$ | $87.4 \pm 1.4$ | $86.8 \pm 2.3$ | $\mathbf{88.5 \pm 0.8}$ |
| GPQA | $39.1 \pm 0.7$ | $38.1 \pm 0.9$ | $38.7 \pm 1.6$ | $38.5 \pm 1.9$ | $\mathbf{40.4 \pm 1.2}$ |
| GSM8K | $94.5 \pm 1.9$ | $93.8 \pm 1.1$ | $94.3 \pm 1.6$ | $93.9 \pm 1.5$ | $\mathbf{95.2 \pm 1.4}$ |

fine-tuning service to obtain a customized model. Fine-tuning is performed using the official OpenAI API[4]. We report the results in **Appendix A.1**.

Due to page limits, additional benchmark results are presented in **Appendix A.1**. Hyperparameter ablation studies are presented in **Appendix A.2**. Illustrative examples are presented in **Appendix A.3**.

### 4.3 MOLECULE OPTIMIZATION

In this part, we apply te proposed UFPL algorithm to a real world drug molecular optimization task. The supervision for each molecule is defined by the optimal counterparts, evaluated based on the Vina score and QED score. We begin with five clinically approved drugs from the dataset as the initial set of "high-confidence" pseudo-supervised data. GPT-4o is used as the LLM, with the prompt text adopted from TextGrad.

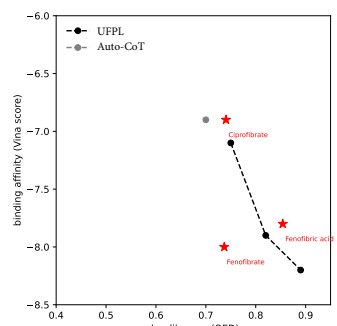

In Figure 5, we present the drug molecules refined by the proposed UFPL in the final three iterations, alongside the molecule refined by Auto-CoT and three clinically approved drugs Ciprofibrate, Fenofibrate, and Fenofibric acid. We observe that the molecule refined by UFPL is structurally close to clinically approved drugs, while achieving better QED and Vina scores and outperforming the Auto-CoT method.

Based on this empirical result, UFPL explores the entire unsupervised dataset to generate more refined outputs, while leveraging the TextGrad framework to produce explainable decisions, which allow researchers to clearly understand how and why a molecule's structure is generated. These results underscore the

**Figure 5:** Vina score and QED score of the molecules refined by UFPL and Auto-CoT compared to clinically approved compounds. The molecule refined by UFPL exhibits greater structural similarity to its closest approved counterpart while achieving better QED and Vina scores.

promising potential of the proposed UFPL algorithm in scientific discovery tasks.

### 5 CONCLUSION

In this paper, we investigate specialization of general purpose LLM without retraining model parameters or relying on human supervision. A straightforward approach is to generate "high-confidence" pseudo-supervised data and then apply in-context learning or prompt learning; however, using the few-shot examples only at inference rather than during prompt learning create a mismatch between how the prompt is learned and how it is used. We propose UFPL, an algorithm that iteratively identifies "high confidence" pseudo supervised data and jointly optimizes the prompt while refining pseudo supervision, thereby specializing general purpose LLMs for downstream tasks. This joint optimization aligns prompt training with usage by requiring the learned prompt to produce consistent answers when pseudo-supervised data from the downstream task are used as few-shot examples. Theoretical analysis shows that, in a simplified multi-class classification setting, UFPL encourages pseudo-supervision to form a low-dimensional structure, helping to mitigate overfitting and improve generalization. Experiments on several benchmarks and a real-world molecule optimization task demonstrate the effectiveness of UFPL.

---

[4]https://platform.openai.com/docs/guides/fine-tuning

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

APPENDIX

# A   ADDITIONAL EXPERIMENTAL RESULTS

## A.1   ADDITIONAL PERFORMANCE COMPARISON ON BENCHMARKS

We first report the performance of each sub-dataset within MMLU in Table 3.

**Table 3:** Performance comparisons across sub-datasets in MMLU with GPT-4o. We report the average accuracy (%) and standard deviation over 5 runs. The best results are in **bold** and (↑ ·) indicates the improvement over Direct in terms of average accuracy.

| Model | Dataset | Direct | ICL | Auto-CoT | USP | SR (BDPL) | SR (RLprompt) | UFPL |
|-------|---------|--------|-----|----------|-----|-----------|---------------|------|
| GPT-4o | MAR | $90.2 \pm 2.0$ | $90.7 \pm 1.7$ | $88.9 \pm 1.7$ | $\mathbf{92.4 \pm 0.9}$ | $91.3 \pm 1.8$ | $91.0 \pm 0.8$ | $92.1 \pm 0.8$ (↑1.9) |
| | MAN | $76.8 \pm 1.4$ | $76.4 \pm 1.0$ | $76.5 \pm 1.0$ | $77.5 \pm 1.6$ | $79.0 \pm 1.2$ | $78.2 \pm 0.9$ | $\mathbf{81.1 \pm 1.4}$ (↑4.3) |
| | HSM | $50.9 \pm 2.9$ | $47.5 \pm 2.2$ | $47.4 \pm 2.2$ | $51.4 \pm 2.3$ | $53.4 \pm 1.8$ | $53.2 \pm 1.1$ | $\mathbf{55.6 \pm 1.6}$ (↑4.7) |
| | HCS | $90.8 \pm 2.7$ | $91.0 \pm 2.1$ | $89.1 \pm 2.1$ | $89.9 \pm 2.3$ | $92.5 \pm 2.1$ | $91.3 \pm 2.2$ | $\mathbf{93.1 \pm 1.4}$ (↑2.3) |
| | CMed | $61.9 \pm 1.8$ | $58.4 \pm 3.4$ | $58.4 \pm 3.4$ | $61.8 \pm 2.1$ | $61.4 \pm 1.7$ | $59.5 \pm 3.0$ | $\mathbf{63.8 \pm 2.3}$ (↑1.9) |
| | CMath | $40.7 \pm 4.2$ | $40.8 \pm 2.5$ | $40.2 \pm 2.5$ | $41.1 \pm 2.8$ | $44.3 \pm 2.7$ | $43.3 \pm 1.3$ | $\mathbf{46.1 \pm 1.6}$ (↑5.4) |
| | CCS | $68.4 \pm 2.4$ | $71.5 \pm 1.3$ | $69.6 \pm 1.3$ | $69.8 \pm 2.3$ | $71.8 \pm 1.8$ | $71.0 \pm 1.6$ | $\mathbf{73.2 \pm 1.0}$ (↑4.8) |
| | AST | $86.6 \pm 2.5$ | $86.8 \pm 2.3$ | $86.5 \pm 2.3$ | $87.1 \pm 2.1$ | $85.6 \pm 3.6$ | $\mathbf{88.0 \pm 2.8}$ | $87.2 \pm 1.5$ (↑0.6) |
| | RND | $68.7 \pm 1.1$ | $68.9 \pm 1.2$ | $68.3 \pm 1.2$ | $70.4 \pm 1.7$ | $70.6 \pm 1.7$ | $70.5 \pm 1.3$ | $\mathbf{72.8 \pm 2.0}$ (↑4.1) |

In certain cases, users may prefer a customized model over refined generation for downstream tasks. To evaluate the performance of the fine-tuned model for both the proposed method and the baselines, we first learn the prompt and pseudo-supervision using 20% of the original dataset. The model is then fine-tuned on this refined dataset and evaluated on the remaining 80% of the data. We report the results in Table 4. Our proposed method consistently outperforms other contenders, indicating higher quality in the refined generation compared to existing approaches.

**Table 4:** Performance comparisons on fine-tuned models across sub-datasets in MMLU with GPT-4o. We report the average accuracy (%) and standard deviation over 5 runs. The best results are in **bold**. The best results are in **bold** and (↑ ·) indicates the improvement over Direct in terms of average accuracy.

| Model | Dataset | Direct | ICL | Auto-CoT | USP | SR (BDPL) | SR (RLprompt) | UFPL |
|-------|---------|--------|-----|----------|-----|-----------|---------------|------|
| GPT-4o | MAR | $91.1 \pm 2.3$ | $88.9 \pm 1.5$ | $89.9 \pm 1.3$ | $92.7 \pm 1.1$ | $91.5 \pm 1.7$ | $92.4 \pm 0.4$ | $\mathbf{93.6 \pm 0.8}$ (↑2.5) |
| | MAN | $76.9 \pm 1.1$ | $77.8 \pm 1.5$ | $77.0 \pm 1.3$ | $78.5 \pm 2.0$ | $79.5 \pm 1.6$ | $79.0 \pm 1.0$ | $\mathbf{82.0 \pm 1.8}$ (↑5.1) |
| | HSM | $51.1 \pm 2.8$ | $47.9 \pm 2.0$ | $47.6 \pm 2.2$ | $52.0 \pm 1.9$ | $54.0 \pm 2.1$ | $53.7 \pm 0.7$ | $\mathbf{56.9 \pm 2.1}$ (↑5.8) |
| | HCS | $91.6 \pm 2.5$ | $89.6 \pm 2.4$ | $89.7 \pm 1.8$ | $91.6 \pm 1.8$ | $93.7 \pm 2.0$ | $92.2 \pm 1.7$ | $\mathbf{94.1 \pm 1.7}$ (↑2.5) |
| | CMed | $62.9 \pm 1.5$ | $59.9 \pm 2.9$ | $59.4 \pm 3.0$ | $62.6 \pm 2.5$ | $62.7 \pm 1.9$ | $61.2 \pm 2.6$ | $\mathbf{64.1 \pm 2.4}$ (↑1.2) |
| | CMath | $41.6 \pm 4.0$ | $41.2 \pm 2.3$ | $41.3 \pm 2.0$ | $42.4 \pm 2.9$ | $45.1 \pm 2.6$ | $44.3 \pm 1.6$ | $\mathbf{47.2 \pm 1.8}$ (↑5.6) |
| | CCS | $69.7 \pm 2.0$ | $70.8 \pm 1.4$ | $70.6 \pm 1.1$ | $70.6 \pm 2.5$ | $72.9 \pm 1.6$ | $72.4 \pm 1.8$ | $\mathbf{74.7 \pm 1.3}$ (↑5.0) |
| | AST | $87.4 \pm 2.6$ | $87.7 \pm 2.4$ | $87.3 \pm 2.1$ | $88.5 \pm 2.1$ | $86.7 \pm 3.1$ | $\mathbf{89.1 \pm 2.4}$ | $88.7 \pm 1.9$ (↑1.3) |
| | RND | $69.4 \pm 1.3$ | $69.3 \pm 1.4$ | $69.5 \pm 1.1$ | $71.3 \pm 1.4$ | $71.9 \pm 1.4$ | $71.6 \pm 1.0$ | $\mathbf{73.8 \pm 1.8}$ (↑4.4) |

## A.2   ABLATION STUDIES

In this part, we conduct ablation studies on the proposed UFPL algorithm, analyzing the impact of generation of "high-confidence" pseudo-supervised data, the selection of in-context examples, the computational overhead of UFPL, and the choice of LLM used in the pipeline.

**Number of in-context demonstrations.**   Finally, we investigate the impact of the number of in-context demonstrations by selecting different numbers of $K$-nearest samples for each query, following the distance metric used in (Liu et al., 2022). The comparison results are reported in Table 5. It can be observed that the UFPL algorithm outperforms both the Direct and ICL methods on nearly all datasets across different values of $K$. This highlights the benefit of leveraging pseudo-supervised data as in-context demonstrations during the prompt optimization phase. Based on our empirical results, setting $K = 5$ is recommended to achieve satisfactory performance.

**Table 5:** Performance comparisons with varying number of in-context examples on benchmark datasets. We report the average accuracy (%) and standard deviation over 5 runs. The best results are in **bold**.

| Method | MNLI | QQP | SST-2 | MRPC | CoLA | WNLI | RTE | RND |
|---|---|---|---|---|---|---|---|---|
| Direct | $91.7 \pm 2.3$ | $71.4 \pm 1.0$ | $89.6 \pm 1.5$ | $90.9 \pm 2.0$ | $69.7 \pm 1.7$ | $90.8 \pm 1.6$ | $92.9 \pm 1.2$ | $68.7 \pm 1.1$ |
| ICL ($k=3$) | $89.3 \pm 1.9$ | $68.5 \pm 2.1$ | $88.9 \pm 2.4$ | $88.3 \pm 1.7$ | $66.4 \pm 2.3$ | $87.5 \pm 1.7$ | $88.3 \pm 1.2$ | $67.5 \pm 1.5$ |
| ICL ($k=5$) | $90.4 \pm 2.0$ | $71.6 \pm 2.0$ | $88.4 \pm 0.7$ | $91.0 \pm 1.5$ | $69.7 \pm 2.3$ | $87.3 \pm 1.7$ | $93.1 \pm 1.0$ | $68.9 \pm 1.2$ |
| UFPL ($k=3$) | $91.5 \pm 2.1$ | $72.5 \pm 2.1$ | $91.3 \pm 1.7$ | $92.3 \pm 1.8$ | $\mathbf{71.8 \pm 1.5}$ | $91.0 \pm 1.7$ | $93.1 \pm 2.0$ | $71.5 \pm 2.6$ |
| UFPL ($k=5$) | $\mathbf{92.0 \pm 1.8}$ | $\mathbf{73.2 \pm 2.0}$ | $\mathbf{92.7 \pm 1.1}$ | $\mathbf{93.4 \pm 1.7}$ | $71.2 \pm 1.1$ | $\mathbf{91.1 \pm 1.4}$ | $\mathbf{94.9 \pm 1.6}$ | $\mathbf{72.8 \pm 2.0}$ |

**Confidence threshold.** We first investigate the confidence threshold $\gamma$ for generating "high-confidence" pseudo-labeled data. Experiments are conducted across both the question answering and natural language inference tasks, using average accuracy as the evaluation metric. The results are presented in Figure 6. We observe that setting the confidence threshold between 0.6 and 0.7 yields stable and satisfactory performance across all experiments. A lower threshold may introduce incorrect pseudo-labels, negatively affecting performance, while a

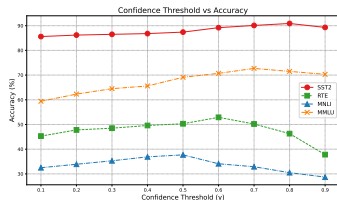

**Figure 6:** Accuracy with different $\gamma$.

higher threshold can limit the amount of selected pseudo-supervised data, also leading to performance degradation. Based on these findings, we recommend setting the confidence threshold in the range of 0.6 to 0.7 for practical applications.

### A.3 ILLUSTRATIVE EXAMPLE

In this section, we present the optimized prompts for the SimpleQA (Wei et al., 2024) dataset as an illustration.

---

**Example of the SimpleQA dataset**

**Prompt at initialization:**
*You will answer a general-knowledge question on \$topic topic. Always conclude the last line of your response should be of the following format: 'Answer: \$VALUE' where VALUE is a \$answer_type value."*

**Prompt refined by UFPL:**
*You will answer a general-knowledge question. Restate the question in your own words to ensure understanding. Compare it with the examples provided above, note any shared entities and relations. Reason through the composition using evidence from both the question and demonstrations. Cross Check your conclusion, ensure it does not contradict any high confidence example. Always conclude the last line of your response should be of the following format: 'Answer: \$VALUE' where VALUE is a \$answer_type value."*

---

## B PROOF OF THEOREM 1

**Theorem 2** (Restatement of Theorem 1). *Under Assumptions 1 and 2, let $S = \{(x_i, \widetilde{y}_i)\}_{i=1}^n$ be the refined pseudo supervision produced by UFPL. Let $f_S : \mathbb{R}^d \to \mathbb{R}^K$ be the ERM score and $h_S(x) = \arg\max_{k \leq K} f_{S,k}(x)$ the induced classifier. Then there exists $\gamma > 0$ such that, for all $i$,*

$$f_{S,\widetilde{y}_i}(x_i) \ - \ \max_{j \neq \widetilde{y}_i} f_{S,j}(x_i) \ \geq \ \gamma.$$

*If, in addition, $f_S$ is L-Lipschitz in $x$, set $r := \gamma/(3L)$. Then:*

*(a) (Local purity) For every $i$ and every $x$ with $\|x - x_i\|_2 < r$, we have $h_S(x) = \widetilde{y}_i$.*

*(b) (Cluster connectivity) Define $\mathcal{U} := \bigcup_{i=1}^n B(x_i, r)$. If two training points $x_i, x_j$ admit a chain of indices $i = i_0, i_1, \ldots, i_m = j$ such that $B(x_{i_t}, r) \cap B(x_{i_{t+1}}, r) \neq \emptyset$ for all $t$, then $\widetilde{y}_i = \widetilde{y}_j$. Equivalently, $h_S$ is constant on each path-connected component of $\mathcal{U}$.*

*Proof.* **Step 1: Uniform margin at training points.** For each $i$, let $f^{(-i)}$ be the leave-one-out ERM score trained on $S^{(-i)}$. By Assumption 1, there exists $\gamma_i > 0$ with

$$f_{\widetilde{y}_i}^{(-i)}(x_i) - \max_{j \neq \widetilde{y}_i} f_j^{(-i)}(x_i) \geq \gamma_i.$$

Assumption 2 yields, for all classes $k$, $\left| f_{S,k}(x_i) - f_k^{(-i)}(x_i) \right| \leq C/n$. Hence, by the triangle inequality,

$$f_{S,\widetilde{y}_i}(x_i) - \max_{j \neq \widetilde{y}_i} f_{S,j}(x_i) \geq \left[ f_{\widetilde{y}_i}^{(-i)}(x_i) - \max_{j \neq \widetilde{y}_i} f_j^{(-i)}(x_i) \right] - \frac{2C}{n}$$

$$\geq \gamma_i - \frac{2C}{n}.$$

Let $\gamma := \min_i \gamma_i - \frac{2C}{n} > 0$ (for sufficiently large $n$). This proves the uniform margin claim.

**Step 2: Local label stability via Lipschitz continuity.** Assume $f_S$ is $L$-Lipschitz: $\|f_S(x) - f_S(x')\|_\infty \leq L\|x - x'\|_2$. Fix any $i$ and any $x$ with $\|x - x_i\|_2 < r = \gamma/(3L)$. Then, for all $k$,

$$\left| f_{S,k}(x) - f_{S,k}(x_i) \right| \leq L\|x - x_i\|_2 < Lr = \gamma/3.$$

Therefore,

$$f_{S,\widetilde{y}_i}(x) \geq f_{S,\widetilde{y}_i}(x_i) - \gamma/3,$$
$$\max_{j \neq \widetilde{y}_i} f_{S,j}(x) \leq \max_{j \neq \widetilde{y}_i} f_{S,j}(x_i) + \gamma/3.$$

Subtracting gives

$$f_{S,\widetilde{y}_i}(x) - \max_{j \neq \widetilde{y}_i} f_{S,j}(x) \geq \left[ f_{S,\widetilde{y}_i}(x_i) - \max_{j \neq \widetilde{y}_i} f_{S,j}(x_i) \right] - \frac{2\gamma}{3} \geq \gamma - \frac{2\gamma}{3} = \frac{\gamma}{3} > 0.$$

Hence $h_S(x) = \widetilde{y}_i$, proving (a). If a deterministic tie-breaking rule is specified, one can take $r = \gamma/(2L)$ with a nonnegative gap.

**Step 3: Propagation along overlapping balls.** Suppose $B(x_{i_t}, r) \cap B(x_{i_{t+1}}, r) \neq \emptyset$. Pick any $z \in B(x_{i_t}, r) \cap B(x_{i_{t+1}}, r)$. By (a), $h_S(z) = \widetilde{y}_{i_t}$ and also $h_S(z) = \widetilde{y}_{i_{t+1}}$, hence $\widetilde{y}_{i_t} = \widetilde{y}_{i_{t+1}}$. By induction along the chain $i = i_0, i_1, \ldots, i_m = j$, we get $\widetilde{y}_i = \widetilde{y}_j$, proving (b). Equivalently, since labels are constant within each $B(x_i, r)$ and agree on overlaps, $h_S$ is constant on every path-connected component of $\mathcal{U} = \bigcup_i B(x_i, r)$. $\square$

## C  IMPLEMENTATION DETAILS

In this section, we present the prompts (manual templates) used by TextGrad for each dataset.

### C.1  PROMPT DESIGN IN TEXTGRAD

For every task we compose a system prompt that fixes the global behaviour of GPT-4o and a task prompt that encodes the input variables.

The forward model receives the concatenation: `<task-prompt>` + `<in-context demos>` + `<query>`.

**Confidence filter.** A sample is kept in the loss only if

$$\max_c p_\theta(y = c \mid x) \geq 0.80$$

This threshold was tuned once on GLUE and reused everywhere else.

**Hyper-parameters.**

- Optimiser: TGD (step size 1.0, temperature 0.7);
- Prompt length cap: 256 GPT-4o tokens;
- Demonstrations per query: $K = 4$;
- PAPO iterations $T$: 10 (classification) / 5 (reasoning datasets).

| Dataset | Initial prompt $z_0$ |
|---|---|
| **SST-2** | Review: {sentence}, Options: {options}. Answer: |
| **CoLA** | Sentence: {sentence} Options: {options}. Answer: |
| **MNLI** | Premise: {premise}\nHypothesis: {hypothesis}\nOptions: {options}. Answer: |
| **QQP** | Question 1: {question1}\nQuestion 2: {question2}\nOptions: {options}. Answer: |
| **MRPC** | Sentence 1: {sentence1}\nSentence 2: {sentence2}\nOptions: {options}. Answer: |
| **RTE** | Premise: {sentence1}\nHypothesis: {sentence2}\nOptions: {options}. Answer: |
| **WNLI** | Sentence 1: {sentence1}\nSentence 2: {sentence2}\nOptions: {options}. Answer: |
| **CAIS/MMLU** | Question: {question}, Options: {options}. Answer: |
| **SimpleQA** | You will answer a general-knowledge question on $topic topic. Always conclude the last line of your response should be of the following format: 'Answer: $VALUE' where VALUE is a $answer_type value." |
| **GPQA** | You will answer a professional knowledge question. Think step-by-step. Always finish with Answer: $OPTION where OPTION is the letter of the correct choice. |

**Table 6:** Initial prompt templates for all datasets evaluated in the paper.

## C.2 PROMPT DESIGN FOR EACH TASK

# D USE OF LARGE LANGUAGE MODELS

We use LLMs to check grammar.

