# OpenReview forum: "Unsupervised Prompt Learning with Few-shot Examples for Answering Objective Questions"
_ICLR.cc/2026/Conference — Submitted to ICLR 2026_

### Official Review · Reviewer_KUt5 · 2025-10-24

**Soundness:** 3
**Presentation:** 2
**Contribution:** 2
**Rating:** 2
**Confidence:** 4

**Summary:**

This paper proposes UFPL, which selects high-confident answers as pseudo labels and uses few-shot demonstrations in the prompt optimization stage.

**Strengths:**

1. This work bridges the gap between prompt tuning and few-shot ICL.

2. The experiment is comprehensive.

**Weaknesses:**

1. This work is incremental to me. The selection of high-confident pseudo labels and the optimization algorithm are from existing works [1,2]. The claimed contribution is adding a demonstration into the prompt tuning process, which only introduces additional information to the input without addressing many technical challenges. Moreover, the authors provide a theoretical analysis in section 3.3 but the analysis is about how pseudo supervision helps optimization instead of supporting the claimed contribution of this work.

2. The figures in this paper is hard to interpret. Is the red D_i in figure 1 in prompt training and prompt using phase representing the same demonstration? The format of the caption in Figure 2 is not very formal. The font used in the Figures for the experiments is too small.

[1] Huang J, Gu S S, Hou L, et al. Large language models can self-improve[J]. arXiv preprint arXiv:2210.11610, 2022.
[2] Yuksekgonul M, Bianchi F, Boen J, et al. Textgrad: Automatic" differentiation" via text[J]. arXiv preprint arXiv:2406.07496, 2024.

**Questions:**

1. This paper uses self-consistency to produce pseudo labels to guide the optimization. Have the authors tried methods to mitigate the consistent error (confident but erroneous response) in this stage? Is there any solution to this for further improving the ICL performance?

---

> ### Author Response · Authors · 2025-11-26
>
> Thank you for your detailed review and thoughtful feedback. **We respectfully believe that some key contributions of our work may have been misunderstood, potentially affecting the evaluation, and we provide clarification below.**
>
> Below are our responses to the specific questions and comments. We will revise the draft accordingly. We are happy to answer any further questions you may have.
>
> ---
>
> We would first like to clarify the novelty and contributions of our work. **At the conceptual level**, prior methods rely on few-shot examples with **fixed** pseudo-supervision, whereas our approach incorporates demonstrations whose pseudo labels are **continuously refined**. This design keeps training tightly aligned with demonstration-conditioned inference and provides a theoretically grounded benefit: the refined labels form a clustered, low-dimensional structure that serves as an implicit regularizer and **enhances generalization** [1].
>
> **At the technical level**, our key contribution is a new unsupervised learning scheme that operates in settings where both the model’s predictions and the pseudo-supervision depend on in-context demonstrations. Specifically, we introduce a **text-valued contrastive loss** that compares demonstration-conditioned pseudo supervision with a reference high-confident prediction. Importantly, this loss is evaluated under the **same** demonstration construction used at inference, where demonstrations are formed via nearest-neighbor retrieval and iteratively refined pseudo labels. This yields **the first practical and fully unsupervised procedure** for updating prompts while ensuring that training remains tightly aligned with demonstration-conditioned inference.
>
> In the following, we answer the questions point by point:
>
> **Q1**: Is the red $D_i$ in figure 1 in prompt training and prompt using phase representing the same demonstration?
>
> **A1**: After optimization, they represent the same demonstrations for the a certain query.
>
> ---
>
> **Q2**: This paper uses self-consistency to produce pseudo labels to guide the optimization. Have the authors tried methods to mitigate the consistent error (confident but erroneous response) in this stage? Is there any solution to this for further improving the ICL performance?
>
> **A2**: In our current framework, self-consistency is used to choose high-confidence pseudo labels, and we have already incorporate two mechanisms to mitigate confident but erroneous outputs: (i) nearest-neighbor-based demonstration selection (expect the initial iteration), which reduces the impact of mislabeled examples by conditioning predictions on semantically similar instances, and (ii) iterative refinement, where pseudo labels are regenerated after each prompt update, allowing incorrect early labels to be corrected over iterations.
>
> Improving robustness to consistent errors is an interesting direction. Techniques such as ensemble-based disagreement filtering, entropy regularization, or lightweight verifier models could be integrated into our pipeline to further enhance pseudo-label reliability and downstream ICL performance. We consider this a promising avenue for future work.
>
> ---
>
> Reference:
>
> [1] Zhu X, Goldberg A. Introduction to semi-supervised learning[M]. Morgan & Claypool Publishers, 2009.

---

> > ### Author Response · Authors · 2025-11-28
> >
> > Dear Reviewer KUt5,
> >
> > Thank you again for taking the time to review our paper and our rebuttal!
> >
> > We would like to kindly ask whether our responses have adequately addressed your main concerns. We would be happy to provide further clarification or additional empirical results if needed.
> >
> > Best regards,
> >
> > Authors

---

### Official Review · Reviewer_fmGa · 2025-10-29

**Soundness:** 2
**Presentation:** 1
**Contribution:** 2
**Rating:** 2
**Confidence:** 4

**Summary:**

The paper introduces Unsupervised Few-Shot Prompt Learning (UFPL), a method to specialize general-purpose LLMs for objective question-answering tasks without parameter updates or human supervision. By jointly optimizing a textual prompt and refining pseudo-supervision from high-confidence in-context examples, UFPL closes the gap between prompt training and inference-time usage. The approach leverages gradient-based textual feedback (via TextGrad) to iteratively improve both prompt and pseudo-labels. A simplified theoretical analysis links UFPL to implicit ERM regularization, and extensive experiments, including a real-world molecule optimization task, demonstrate strong gains over baselines.

**Strengths:**

- The joint optimization of prompt and pseudo-supervision with aligned training-inference objectives is novel and addresses a clear practical pain point: the train-test mismatch in prior high-confidence pseudo-labeling pipelines.
- Translating gradients into textual critiques enables prompt refinement without access to model parameters, a clean, scalable mechanism.
- Experiment strengths:
    + Broad coverage across LLMs and standard QA benchmarks, plus a challenging real-world molecule optimization task.
    + Clear ablation studies and comparisons to recent unsupervised/zero-shot baselines.

**Weaknesses:**

- Unclear Learning Objective in Eq. (4): Eq. (4) describes learning objectives based on high-confidence pseudo-supervised data.
However, the loss term encourages the model with optimized prompt $z$ to match the zero-shot
baseline ($\mathbf{z}_0, \emptyset$). This seems counterintuitive: we often expect specialization to
diverge from zero-shot behavior on selected high-confidence inputs, not converge to it.
- The theoretical analysis rests on Assumption 1 (Leave-one-out correctness) and
Assumption 2 (Uniform stability), both of which appear highly idealized:
  + Assumption 1 requires that for every $i$, $h^{(-i)}(x_i) = y_i$, i.e., the demonstrator
perfectly recovers the true label when excluding $x_i$. This is unlikely in low-data, noisy,
or ambiguous settings.
  + Assumption 2 imposes Lipschitz continuity and bounded deviation of the score function
under leave-one-out, again, strong for real LLMs.
- Moreover, Theorem 1 concludes that UFPL induces classifier separation $\geq \gamma$, but:
   + It is unclear how these assumptions reflect the unique properties of UFPL vs. any ICL-based
method.
    + No empirical validation of these assumptions is provided (e.g., measuring leave-one-out
accuracy or stability).
- The set of baselines is inadequate and seems to be chosen arbitrarily. Please consider recent and stronger prompt optimization approaches. For example, RL-based [1], supervised method [2], and LLM-based [3] methods. At least, please discuss related recent works and explain why your method is different and/or cannot be compared with them.

[1]  Do et al. "Large Language Model Prompting with Episodic Memory." In ECAI. 2024.

[2] Do, Viet-Tung, Xuan-Quang Nguyen, Van-Khanh Hoang, Duy-Hung Nguyen, Shahab Sabahi, Jeff Yang, Hajime Hotta, Minh-Tien Nguyen, and Hung Le. "Automatic prompt selection for large language models." In Pacific-Asia Conference on Knowledge Discovery and Data Mining, pp. 91-102. Springer, Singapore, 2025.

[3] Yang, C., et al.: Large language models as optimizers. arXiv preprint arXiv:2309.03409 (2023)

**Questions:**

Please clarify the following issues:
- Please explain the motivation for Eq. (4)
-  Notation: D, D_k, D_l, D_i, \mathcal{D} used interchangeably (esp. Between Eq. 5 and 6)
- S_l: unclear if S_l is a set of indices or inputs x_k
- L93: BDPL is undefined
- L96: TextGrad might not rely on human supervision
- Algorithm 1: Key formulas (e.g., gradient $\frac{\partial L}{\partial z}$, update rule) missing; parameters like $m$, $T$, $\gamma$ appear but are not being used in pseudocode.
- Table 1: PAPO is undefined.
- Other format error: - L218: Zhao et al. (2021) instead of (Zhao et al. 2021).

---

> ### Author Response · Authors · 2025-11-26
> **Responses to Reviewer fmGa (1/2)**
>
> Thank you for your detailed review and thoughtful feedback. **We respectfully believe that some key contributions of our work may have been misunderstood, potentially affecting the evaluation, and we provide clarification below.**
>
> Below are our responses to the specific questions and comments. We will revise the draft accordingly. We are happy to answer any further questions you may have.
>
> ---
>
> We first clarify Question 1 regarding our learning objective, and then explain our theoretical analysis and the rationale behind our choice of baselines. We subsequently clarify the relationship between our method and the ``recent and stronger prompt optimization approaches'' mentioned by the reviewer.
>
> **Q1**: Please explain the motivation for Eq. (4)
>
> **A1**: We clarify that Eq. (4) is **not** our proposed learning objective but a formalization of prior pseudo-label-based prompt learning methods. It serves as a conceptual baseline objective to **illustrate the limitation of existing prompt learning approaches**, which optimize prompts using high-confidence pseudo-supervised data without considering demonstration-conditioned usage at inference.
>
> In the paragraph immediately following Eq. (4), we explicitly point out the limitation of this objective, and this motivates our optimization objective in Eq. (5). **Our method is based on Eq. (5), which does not encourage convergence to the zero-shot baseline.**
>
> As we stated in Line 226, the theoretical analysis is standard and intended to support the approach, not to claim a theoretical contribution. Our optimization objective encourages the pseudo supervision to form a clustered structure in the output space. Consequently, queries with similar semantics are encouraged to receive the same refined label, helping mitigate overfitting and improve generalization.
>
> With this clarification, we note that the three suggested baselines [1,2,3] all require ground-truth labels or accuracy-based rewards, which makes them incompatible with our fully unsupervised setting. The feasible way to use them is to treat high-confidence pseudo labels as surrogate ground truth and apply these methods in a supervised manner, which corresponds to the optimization in Eq. (4). We test them under this setup and added the results and discussion in the revised manuscript, including a clarification in related work.
>
> Moreover, the used baselines follow a simple principle: they should operate in the same fully unsupervised objective-QA setting as UFPL, where no ground-truth labels or reward models, and supervision comes solely from LLM-generated pseudo labels. Therefore, we use only methods such as ICL variants, pseudo-supervised prompt learning, and self-consistency refinement that function without labeled data. The three suggested baselines [1,2,3] also belong to pseudo-supervised prompt learning.
>
> |Model|Dataset|Direct|Auto-CoT|POEM[1]|APS[2]|SR(RLprompt)|UFPL|
> |---------|-------------|----------------------|-----------------------|------|--------|------------------------|-----------------------------------------|
> |**GPT-4o**|GPQA|47.4±0.2|47.9±0.7|48.2±0.4|47.9±1.2|48.1±1.9|**49.7±1.5**(↑2.3)|
> ||SimpleQA|38.2±0.8|38.9±1.0|38.8±1.5|38.3±0.9|37.4±1.1|**39.6±0.9**(↑1.4)|
> ||TruthfulQA|71.8±0.5|72.1±1.2|73.0±1.9|72.6±1.3|72.3±1.0|**74.3±1.2**(↑2.5)|
> ||MMLU|87.5±0.3|88.1±1.3|88.6±1.7|88.0±1.1|89.2±1.7|**90.4±1.9**(↑2.9)|
> ||GSM8k|93.9±0.5|94.3±0.9|95.0±1.1|94.6±0.7|94.9±1.5|**95.7±1.3**(↑1.8)|
> ||HellaSwag|94.7±0.4|95.1±0.6|95.3±0.9|95.2±1.2|95.5±0.8|**96.3±0.5**(↑1.6)|
> ||BBH|83.1±0.8|83.3±1.1|84.9±1.2|84.7±1.6|85.1±1.2|**86.2±0.8**(↑3.1)|
> |**Qwen3**|GPQA|47.9±1.3|48.4±0.5|48.8±1.7|48.2±1.5|47.5±0.9|**49.9±0.6**(↑2.0)|
> ||SimpleQA|39.1±0.7|39.4±0.9|40.5±2.0|39.6±1.3|40.7±0.9|**41.5±1.1**(↑2.4)|
> ||TruthfulQA|72.9±0.9|73.3±1.2|74.2±1.9|73.8±1.7|74.8±1.2|**75.3±1.1**(↑2.4)|
> ||MMLU|85.3±0.5|86.1±1.2|87.3±1.6|86.6±1.9|86.9±1.6|**88.1±0.7**(↑2.8)|
> ||GSM8k|94.4±1.8|94.9±1.6|95.0±1.1|94.8±1.6|94.5±1.2|**95.9±1.3**(↑1.5)|
> ||HellaSwag|95.1±0.9|95.6±0.8|95.9±1.1|95.3±1.2|96.2±0.9|**96.7±0.8**(↑1.6)|
> ||BBH|87.5±1.1|88.0±1.3|88.2±1.6|87.9±1.5|88.6±1.3|**88.9±1.4**(↑1.4)|
> |**Llama**|GPQA|25.9±0.4|27.1±1.8|27.5±1.9|26.8±2.2|27.8±1.9|**29.3±1.7**(↑3.4)|
> ||SimpleQA|15.3±0.9|15.8±1.1|16.1±1.7|16.3±1.5|16.5±1.6|**16.9±2.1**(↑1.6)|
> ||TruthfulQA|31.9±0.7|32.3±1.0|33.0±1.4|32.3±1.9|32.9±1.4|**33.1±1.5**(↑1.2)|
> ||MMLU|49.1±0.3|49.8±1.4|50.0±1.7|50.2±1.2|50.5±1.5|**52.8±0.9**(↑3.7)|
> ||GSM8k|44.3±0.4|45.3±1.9|46.2±0.9|45.9±1.3|46.3±1.7|**47.5±1.3**(↑3.2)|
> ||HellaSwag|41.3±0.6|42.1±0.9|43.5±1.1|42.7±1.3|44.0±0.9|**44.7±0.6**(↑3.4)|
> ||BBH|37.6±1.1|38.6±1.3|39.1±1.9|39.3±1.7|39.8±1.3|**40.3±1.4**(↑2.7)|

---

> > ### Author Response · Authors · 2025-11-26
> > **Responses to Reviewer fmGa (2/2)**
> >
> > **Q2**: Notation: $D$, $D_k$, $D_l$, $D_i$, $\mathcal{D}$ used interchangeably (esp. Between Eq. 5 and 6)
> >
> > **A2**: Thank you for pointing this out. To clarify, $D$ denotes a general demonstration set, while $D_k$, $D_l$, $D_i$ refer to the specific demonstration sets associated with $x_k$, $x_l$, $x_i$, respectively. The use of $\mathcal{D}$ in line 208 was a typo and should simply be $D$; we have corrected this in the revised manuscript.
> >
> > ---
> >
> > **Q3**: $S_l$: unclear if $S_l$ is a set of indices or inputs $x_k$
> >
> > **A3**: Thank you for the comment. In Eq. (6), $S_l$ is intended to denote the **set of indices** of the selected nearest neighbors, and the corresponding inputs are $\{x_k \mid k \in S_l\}$. We have revised the notation of $D_L$ in line 208 to avoid confusion between indices and inputs.
> >
> > ---
> >
> > **Q4**: L93: BDPL is undefined
> >
> > **A4**: Thank you for the comment. The reference for BDPL was included at the end of that sentence.
> >
> > ---
> >
> > **Q5**: L96: TextGrad might not rely on human supervision
> >
> > **A5**: Thank you for pointing this out. TextGrad does rely on human supervision in certain tasks. In our context, we were specifically referring to its use in objective QA settings. We have revised the statement to make this distinction clear and improve accuracy.
> >
> > ---
> >
> > **Q6**: Algorithm 1: Key formulas (e.g., gradient, update rule) missing; parameters like appear but are not being used in pseudocode.
> >
> > **A6**: Thank you for the comment. The gradients, update rule, and related parameters were already specified in the equations referenced by the pseudocode, so they were not missing. In the pseudocode, we omitted repeating these expressions to avoid redundancy and to keep the main text concise.
> >
> > ---
> >
> > **Q7**: Table 1: PAPO is undefined.
> >
> > **A7**: Thank you for catching this. ``PAPO'' was a typo, it refers to our UFPL method. We have corrected it in the revised manuscript.
> >
> > ---
> >
> > **Q8**: Other format error: - L218: Zhao et al. (2021) instead of (Zhao et al. 2021).
> >
> > **A8**: Thank you for pointing this out. We will revise it accordingly.
> >
> > ---
> >
> > Reference:
> >
> > [1] Do et al. "Large Language Model Prompting with Episodic Memory." In ECAI. 2024.
> >
> > [2] Do, Viet-Tung, Xuan-Quang Nguyen, Van-Khanh Hoang, Duy-Hung Nguyen, Shahab Sabahi, Jeff Yang, Hajime Hotta, Minh-Tien Nguyen, and Hung Le. "Automatic prompt selection for large language models." In PAKDD, 2025.
> >
> > [3] Yang, C., et al.: Large language models as optimizers. arXiv preprint arXiv:2309.03409 (2023)

---

> > > ### Author Response · Authors · 2025-11-28
> > >
> > > Dear Reviewer fmGa,
> > >
> > > Thank you again for taking the time to review our paper and our rebuttal!
> > >
> > > We would like to kindly ask whether our responses have adequately addressed your main concerns. We would be happy to provide further clarification or additional empirical results if needed.
> > >
> > > Best regards,
> > >
> > > Authors

---

### Official Review · Reviewer_saUT · 2025-11-01

**Soundness:** 2
**Presentation:** 3
**Contribution:** 3
**Rating:** 6
**Confidence:** 1

**Summary:**

This paper proposes unsupervised few-shot prompt learning (UFPL) to jointly learn the prompt and refine the overall pseudo-supervision. Both theoretical analysis and empirical results are presented.

**Strengths:**

S1. Well-motivated work

S2. Extensive experiments.

S3. A theoretical analysis, though in a simplified setting.

**Weaknesses:**

W1. UFPL is inherently more expensive and complex than baselines, as shown in Figure 3 and Figure 4, limiting their use in a wider range of scenarios.

W2. There are several hyperparameters, which may be hard to set in a truly unsupervised setting. I did not find hyperparameter analysis in Appendix C. I only saw some limited analysis on one hyperparameter in  question answering and natural language inference tasks in Figure 6. Not sure whether such a conclusion drawn from limited tasks can be generalized.

**Questions:**

Q1. Could you quantitatively compare the computational cost of your approach with the baselines? The current description about Figure 3 and Figure 4 is vague (e.g., 'modest increase').

Q2. Could you conduct hyperparameter analysis on all datasets as in your main experiments?

---

> ### Author Response · Authors · 2025-11-26
>
> Thank you for your detailed review and thoughtful feedback.
>
> Below are our responses to the specific questions and comments. We will revise the draft accordingly. We are happy to answer any further questions you may have.
>
> ---
>
> **Q1**: Could you quantitatively compare the computational cost of your approach with the baselines? The current description about Figure 3 and Figure 4 is t (e.g., 'modest increase').
>
> **A1**: Our computational cost comparison is presented quantitatively in Figures 3 and 4. Below, we summarize the key numbers:
>
> * **Llama-3.2-1B (Open Source Models):**
>   UFPL achieves **1.6–3.4% higher accuracy** than Direct, USP, and RLPrompt, while adding only **approximately 1–2 USD per 1k tokens** of additional cost (Figure 3).
>
> * **Comparison with SFT-LoRA:**
>   UFPL attains higher accuracy at **lower total cost**, since LoRA-based fine-tuning requires **multiple GPU-hours**, making it substantially more expensive than UFPL for comparable improvements (Figure 3).
>
> * **GPT-4o (API Models):**
>   UFPL yields **1–3% accuracy gains** with only **6–12 minutes of overhead per 1k tokens** (Figure 4).
>
> * **Overall Efficiency:**
>   Most of UFPL’s improvements arise in the **second iteration**, indicating favorable cost–performance efficiency relative to baseline methods.
>
> ---
>
> **Q2**: Could you conduct hyperparameter analysis on all datasets as in your main experiments?
>
> **A2**: We conducted hyperparameter analysis across all datasets used in our main experiments. The results are summarized below. As shown below, UFPL consistently improves performance over Direct prompting, and varying the number of retrieved examples leads to stable gains across tasks. We can also find that performance is relatively insensitive to the choice of the confidence threshold, and setting it in the range 0.5–0.7 yields good performance across datasets.
>
> | Method    | GPQA           | SimpleQA       | TruthfulQA     | MMLU           | GSM8K          | HellaSwag      | BBH            |
> | --------- | -------------- | -------------- | -------------- | -------------- | -------------- | -------------- | -------------- |
> | Direct    | 47.4 ± 0.2     | 38.2 ± 0.8     | 71.8 ± 0.5     | 87.5 ± 0.3     | 93.9 ± 0.5     | 94.7 ± 0.4     | 83.1 ± 0.8     |
> | UFPL(k=3) | **49.7 ± 1.5** | 39.2 ± 0.5     | 73.7±0.8       | 89.7±1.6       | 95.3±0.8       | **96.3 ± 0.5** | 85.8±1.1       |
> | UFPL(k=5) | 49.2±1.3       | **39.6 ± 0.9** | **74.3 ± 1.2** | **90.4 ± 1.9** | **95.7 ± 1.3** | 96.1±0.8       | **86.2 ± 0.8** |
>
> | Confidence Threshold   | 0.1     | 0.2     | 0.3     | 0.4     | 0.5     | 0.6     | 0.7     | 0.8     | 0.9     |
> |-----------|-------|-------|-------|-------|-------|-------|-------|-------|-------|
> | GPQA      | 45.3  | 45.8  | 46.1  | 46.3  | 46.7  | 47.2  | 48.3  | 49.7  | 48.5  |
> | SimpleQA  | 36.7  | 36.9  | 38.3  | 38.9  | 39.3  | 39.6  | 39.1  | 38.9  | 38.7  |
> | TruthfulQA| 69.5  | 70.6  | 71.4  | 72.3  | 73.8  | 74.3  | 74.1  | 73.8  | 73.3  |
> | MMLU      | 88.3  | 88.5  | 88.6  | 88.9  | 89.4  | 89.8  | 90.1  | 90.3  | 90.4  |
> | GSM8K     | 92.1  | 92.8  | 93.3  | 93.7  | 94.2  | 94.8  | 95.4  | 95.7  | 95.3  |
> | HellaSwag | 92.7  | 93.5  | 93.8  | 94.2  | 94.7  | 95.3  | 95.8  | 96.3  | 96.0  |
> | BBH       | 83.5  | 83.9  | 84.5  | 84.9  | 85.3  | 85.8  | 86.2  | 85.9  | 85.4  |

---

> > ### Comment · Reviewer_saUT · 2025-11-27
> >
> > Thank you for your responses.
> >
> > Based on your quantitative comparison, I maintain my concern related to W1.
> >
> > For W2, thanks for your results. I has mostly addressed my concern. However, I am not an expert in this particular research problem. After reading the other reviews, I believe my previous scores are already high.

---

> > > ### Author Response · Authors · 2025-11-28
> > >
> > > Dear Reviewer saUT,
> > >
> > > Thank you again for taking the time to review our paper and our rebuttal. We are glad that our additional results on W2 have mostly addressed your concern.
> > >
> > > Regarding W1, a key advantage of our method is that it achieves accuracy gains with *consistently modest extra cost*. Even a one to three percent improvement can lead to noticeably more correct predictions at scale, while the extra computational cost of our method remains small and stable.
> > >
> > > We would be happy to provide further clarification or additional empirical results if needed.
> > >
> > > Thank you!
> > >
> > > Best regards,
> > >
> > > Authors

---

### Official Review · Reviewer_66Hq · 2025-11-01

**Soundness:** 3
**Presentation:** 3
**Contribution:** 3
**Rating:** 6
**Confidence:** 5

**Summary:**

The authors propose to specialize general-purpose large language models (LLMs) for objective question answering tasks without human supervision or parameter updates, which involves queries with a single verifiable answer. Specifically, they aim to address limitations in existing methods like fine-tuning and supervised in-context learning by generating high-confidence pseudo-supervised data from unlabeled downstream tasks using self-consistency chain-of-thought (CoT) reasoning. Experiments are conducted on several benchmark datasets when the experimental results show the consistency accuracy gains.

**Strengths:**

* The proposed method aligns training and inference. By jointly optimizing prompts and pseudo-supervision, it avoids mismatches in conventional approaches and ensures few-shot examples to be used consistently.
* The proposed method relies solely on LLM-generated pseudo-labels from unlabeled data, reducing data collection burdens while achieving high-quality refinements applicable to black-box models.
* The authors also provide theoretical analysis showing implicit regularization for generalization when the method is validated on diverse benchmarks.

**Weaknesses:**

* Iterative refinements and initial distance matrix for example selection add runtime and cost, potentially limiting scalability for large dataset as computational overhead.
* The proposed method relies on self-consistency CoT, so inaccuracies in weak base LLMs could propagate errors.
* The proposed method is limited to objective QA with verifiable answers, so it may not generalize to open-ended or subjective tasks without modifications.

**Questions:**

* How does the proposed method perform on non-objective tasks (e.g., open-ended or creative writing tasks)?
* What are the specific computational environment used in the experiments (e.g., GPU and memory)?
* How does the proposed method handle noisy or ambiguous unlabeled data in the downstream task?
* How does the proposed method perform when applied to multilingual or cross-lingual objective QA tasks?

---

> ### Author Response · Authors · 2025-11-26
>
> Thank you for your detailed review and thoughtful feedback.
>
> Below are our responses to the specific questions and comments. We will revise the draft accordingly. We are happy to answer any further questions you may have.
>
> ---
>
> **Q1**: How does the proposed method perform on non-objective tasks (e.g., open-ended or creative writing tasks)?
>
> **A1**: We evaluate the proposed method on LitBench [1], with detailed results provided below. These experiments show that the UFPL algorithm also achieves satisfactory performance.
>
> In addition, following a related suggestion from another reviewer, we included two extra baselines [3,4] in this new experiment. UFPL remains competitive and performs better than the added baselines.
>
> | Model  | Dataset  | Direct   | Auto-CoT | POEM [3] |  APS [4]  | SR (RLprompt) | UFPL     |
> | ------ | -------- | -------- | -------- | ---- | ----- | ------------- | -------- |
> | GPT-4o | LitBench | 70.3±0.5 | 70.7±0.4 |  71.6 ±0.8   |  70.9  ± 1.1  | 71.5±0.8      | 72.1±1.3 |
> | Qwen3  | LitBench | 68.3±0.9 | 69.1±0.7 |   69.5 ± 0.6  |  69.1 ± 1.0    | 69.2±0.5      | 70.3±0.9 |
> | Llama  | LitBench | 45.7±1.3 | 46.3±0.8 |  47.3 ±1.4   |    46.9±0.9   | 47.4±1.3      | 48.8±2.1 |
>
> ---
>
> **Q2**: What are the specific computational environment used in the experiments (e.g., GPU and memory)?
>
> **A2**: We conducted all experiments on a cluster equipped with NVIDIA A100 80GB GPUs. For open-source models such as Qwen3 and Llama, we used nodes with 8×A100 80GB GPUs. Experiments involving GPT-4o were carried out via API without local computation.
>
> ---
>
> **Q3**: How does the proposed method handle noisy or ambiguous unlabeled data in the downstream task?
>
> **A3**: Currently, UFPL does not explicitly address noisy or ambiguous unlabeled inputs. This is an interesting direction, and several extensions could be incorporated into our framework:
> 1. Before constructing demonstrations, we can apply an embedding-space density estimator to identify outlier or noisy inputs and exclude them from the learning procedure.
> 2. For each input, we can query the LLM multiple times and remove samples whose predicted reasoning traces or extracted entities exhibit high internal inconsistency—an indicator of ambiguity in the input itself.
>
> These extensions integrate naturally into our pipeline and would make the proposed method more robust to noisy or ambiguous unlabeled data. We consider this a promising direction for future work.
>
> ---
>
> **Q4**: How does the proposed method perform when applied to multilingual or cross-lingual objective QA tasks?
>
> **A4**: In this paper, we focus on objective QA tasks with discrete final answers (multiple‑choice or short-form) and evaluate models by answer‑level accuracy on benchmarks such as MMLU, GPQA, SimpleQA, TruthfulQA, GSM8K, HellaSwag, and BBH. In contrast, multilingual benchmarks like MLQA [2] are formulated as extractive span‑prediction tasks and are typically evaluated with Exact Match (EM) and token‑level F1, which requires additional design choices for span extraction and multilingual alignment. Extending UFPL from accuracy‑based objective QA to multilingual extractive QA (e.g., MLQA-style EM/F1 evaluation) is therefore non‑trivial and beyond the scope of this work, and we regard a systematic study in that setting as an important future work.
>
> ---
>
> Reference:
>
> [1] Fein, Daniel, et al. "LitBench: A Benchmark and Dataset for Reliable Evaluation of Creative Writing." arXiv preprint arXiv:2507.00769 (2025).
>
> [2] Lewis, Patrick, et al. "MLQA: Evaluating cross-lingual extractive question answering." In ACL, 2020.
>
> [3] Do et al. "Large Language Model Prompting with Episodic Memory." In ECAI, 2024.
>
> [4] Do, Viet-Tung, Xuan-Quang Nguyen, Van-Khanh Hoang, Duy-Hung Nguyen, Shahab Sabahi, Jeff Yang, Hajime Hotta, Minh-Tien Nguyen, and Hung Le. "Automatic prompt selection for large language models." In PAKDD, 2025.

---

> > ### Author Response · Authors · 2025-11-28
> >
> > Dear Reviewer 66Hq,
> >
> > Thank you again for taking the time to review our paper and our rebuttal!
> >
> > We would like to kindly ask whether our responses have adequately addressed your main concerns. We would be happy to provide further clarification or additional empirical results if needed.
> >
> > Best regards,
> >
> > Authors

---

### Meta-Review · Area_Chair_THDA · 2026-01-01

**Summary:**

Major concerns include incremental novelty, idealized assumption for theoretical analysis and insufficient technical details. The authors propose aligning training and inference with unsupervised few-shot prompt learning. While the reviewer argues that the core components, e.g., high-confidence pseudo-label selection and the optimization strategy, are largely adapted from prior works. The theoretical analysis has quite strong and impractical assumptions. It also lacks empirical validation. Additionally, there are criticisms regarding the unclear learning objective and notion, as well as missing analysis experiments, etc.

**Reviewer Concerns:**

Reviewer-66Hq questions the applicability to open-ended and subjective tasks beyond just objective QA, computational environments, the authors’ applicability to multilingual and cross-lingual objective QA tasks, sensitivity to the “weak”-version of LLMs, etc. In the rebuttal, the authors provide details including the results on LitBench, details regarding the computational environment, and arguments regarding the scope not covering multilingual and cross-lingual work, etc. However, there are more diverse tasks that the authors want to show effectiveness on in the paper. Only partial questions/concerns have been addressed, in belief.

Reviewer-saUT noted that the reviewing confidence score is just 1. In the weakness section, they asked questions about computational cost and hyperparameter analysis. The authors address the concerns in the rebuttal. While the reviewer acknowledges the rebuttal, they remain with the original score.

Reviewer-fmGa raises several points, including unclear motivation for Eq. 4, idealized assumptions for the theoretical analysis, missing stronger baselines, etc. In the rebuttal, the authors clarify that Eq. 4 is not the proposed learning objective but Eq. 5. The authors also provide the requested comparison results. While the response regarding the limiting assumptions (Assumption 1 and Assumption 2) may not fully satisfy the reviewer’s criticism.

Reviewer-KUt5 has concerns mainly on novelty, and also has questions on figure clarity and other methods to mitigate consistent errors instead of self-consistency for producing pseudo-labels. In the rebuttal, the authors clarify the questions on figures and the rationale for using self-consistency. Regarding the novelty part, the authors provide explanations at both the conceptual and technical levels. However, the amount of novelty, from the reviewer’s point of view, may not be sufficient for ICLR.

**Reviewer Scores:**

Reviewer-66Hq and Reviewer-saUT have two positive scores (6, 6). With the rebuttal and my observations shown above, I think these reviewers may keep their scores. Reviewer-fmGa and Reviewer-KUt5 have two negative scores (2, 2). Based on my observations, I think Reviewer-fmGa may maintain the score (or raise it to 4), given that there are still critical concerns regarding the assumptions in the theoretical analysis. Similarly, Reviewer-KUt5 may raise the score to 4, given the concerns about limited novelty.

With all these considerations, given that Reviewer-saUT has a confidence score of 1, I am leaning to believe that this work is not ready for ICLR publication.

---

### Decision · Program_Chairs · 2026-01-26

Reject